# Socioeconomic Disadvantage across the Life Course Is Associated with Diet Quality in Young Adulthood

**DOI:** 10.3390/nu11020242

**Published:** 2019-01-22

**Authors:** Erin L. Faught, Lindsay McLaren, Sharon I. Kirkpatrick, David Hammond, Leia M. Minaker, Kim D. Raine, Dana Lee Olstad

**Affiliations:** 1Department of Community Health Sciences, Cumming School of Medicine, University of Calgary, Teaching, Research, and Wellness Building, 3280 Hospital Drive NW, Calgary, AB T2N 4Z6, Canada; erin.faught@ucalgary.ca (E.L.F.); lmclaren@ucalgary.ca (L.M.); 2School of Public Health and Health Systems, University of Waterloo, 200 University Avenue West, Waterloo, ON N2L 3G1, Canada; sharon.kirkpatrick@uwaterloo.ca (S.I.K.); dhammond@uwaterloo.ca (D.H.); 3School of Planning, University of Waterloo, 200 University Ave, Waterloo, ON N2L 3G1, Canada; lminaker@uwaterloo.ca; 4School of Public Health, University of Alberta, University of Alberta, 4-077 Edmonton Clinic Health Academy, 11405-87 Ave, Edmonton, AB T6G 1C9, Canada; kim.raine@ualberta.ca

**Keywords:** life course theory healthy eating index, socioeconomic position, nutrition, dietary inequities, socioeconomic inequities

## Abstract

Socioeconomic position (SEP) is a key determinant of diet quality across the life course. Young adulthood is a unique period of transition where dietary inequities between groups with lower and higher SEP may widen. This study investigated associations between SEP in both childhood and young adulthood and diet quality in young adulthood. Data from 1949 Canadian young adults aged 18–30 who participated in the Canada Food Study were analyzed. Healthy Eating Index–2015 (HEI-2015) scores were calculated based on one 24-hour dietary recall. Childhood and young adult SEP were represented by self-report of participants’ parent(s)’ and their own highest educational level, respectively. Linear regression was used to examine associations between childhood and adult SEP and adult HEI-2015 score. Mediation analyses examined whether adult SEP mediated the relationship between childhood SEP and adult HEI-2015 score. Lower SEPs in childhood and adulthood were each associated with lower HEI-2015 scores in young adulthood. Adult SEP mediated up to 13.0% of the association between childhood SEP and adult HEI-2015 scores. Study findings provide support for key life course hypotheses and suggest latent, pathway, and cumulative effects of SEP across the early life course in shaping the socioeconomic patterning of diet quality in young adulthood.

## 1. Introduction

Life course perspectives in epidemiology consider pathways through which aspects of socioeconomic position (SEP) independently and cumulatively shape health at all stages of development, and their implications for the creation, maintenance, and reproduction of health inequities both within and between societies [1,2]. Within the life course perspective, three principal hypotheses have been advanced to explain how health inequities are generated, reproduced, and maintained across the life course. These hypotheses are not mutually exclusive; rather, there is strong interdependence between them [3]. The critical periods hypothesis posits that adverse exposures during particular developmental windows become embedded in individuals’ bodies, shaping their developmental capacity and health and social outcomes, independently of subsequent events [4]. The cumulative effects hypothesis (or accumulation of risks) is based on the premise that the duration and intensity of adverse exposures are most relevant to subsequent health [1,2,5]. The pathways hypothesis posits that the effect of disadvantage is indirect by setting individuals on adverse social trajectories that continue across the life course, where one negative factor increases the possibility of experiencing the same negative factor at a subsequent time point (i.e., low childhood SEP increases the possibility of low adult SEP) [2,6].

Life course hypotheses have important implications for the development of inequities in dietary patterns and consequently, diet-related diseases. Diet quality has been shown to differ substantially across SEP groups [7,8], and a longitudinal study has demonstrated that inequities in diet quality widen across the life course, particularly through young adulthood [9]. Consequent diet-related disease patterns, including those for obesity, cardiovascular disease, and diabetes reflect, in part, these dietary inequities [10,11]. There are theoretical reasons to expect that all three life course hypotheses may be implicated in the socioeconomic patterning of diet quality, however, these mechanisms have rarely been studied. In one study by Watts et al. [12], childhood SEP was independently associated with breakfast-skipping and fast food consumption in young adulthood; however, higher SEP in young adulthood partly mitigated the negative impact of a low SEP background on diet. These findings therefore provided support for the critical periods (i.e., enduring independent effect of childhood SEP) and cumulative effects (i.e., partial attenuation of the association between childhood SEP and adult dietary behaviours by SEP in young adulthood) hypotheses in shaping aspects of young adults’ dietary intake. Other evidence indicates that dietary patterns established in childhood tend to endure into adolescence and young adulthood, which is also consistent with life course perspectives [13]. 

Young adulthood has recently been identified as a critical period for the development of less healthful behaviours, including poor dietary intake, meriting further attention to this population group [12,14]. Previous research has demonstrated that adolescents with a low SEP tend to experience greater increases in fast food intake over time compared to their higher SEP peers and have a higher likelihood of being affected by overweight in young adulthood [12]. The multitude of life transitions experienced in this period may contribute to this decline in healthful eating patterns [15], and varying SEP groups may experience these transitions differently. 

Diet has conventionally been assessed in a reductionist manner for the purposes of epidemiological study, considering the contribution of single nutrients or foods to health status rather than considering entire patterns of intake [13,16,17]. Recent developments in nutrition research suggest that dietary patterns may better explain diet-disease relationships than do approaches that focus on particular nutrients, foods, or other dietary constituents [17]. As such, it is recommended that measures of overall diet quality be prioritized in nutritional epidemiology. 

Given gaps in the literature concerning the relevance of life course hypotheses to the development of dietary inequities, the objectives of this study were to better understand the relevance of life course hypotheses to the development of inequities in diet quality in young adulthood. Specifically, we examined: (1) independent and joint associations between SEP in childhood and young adulthood, with diet quality in young adulthood, and (2) whether adult SEP mediated associations between childhood SEP and young adult diet quality. 

We anticipated that findings would provide evidence in support of all three aforementioned life course hypotheses. First, we hypothesized that childhood SEP would be independently associated with young adult diet quality, demonstrating that the impact of childhood socioeconomic conditions on diet quality endures into young adulthood. This would provide support for the critical periods hypothesis. We also hypothesized that childhood and young adult SEP would each be independently associated with diet quality in young adulthood which we interpreted as providing evidence in support of the cumulative effects hypothesis as this suggests that repeated exposure to low SEP leads to poor dietary outcomes [6], and that the effects of SEP would be most pronounced for those with the lowest educational attainment (suggesting greater intensity of negative exposures). Finally, we expected that adult SEP would partially mediate associations between childhood SEP and diet quality in young adulthood, thereby demonstrating the relevance of the pathways hypothesis to the socioeconomic patterning of diet quality. To our knowledge, this is the first study to examine the socioeconomic patterning of diet quality specifically with respect to these three life course hypotheses. Previous works that have investigated the relevance of these hypotheses to dietary inequities have focused on specific nutrients, foods, or diet-related behaviours (e.g., breakfast consumption) as outcomes, or have only addressed the relevance of one life course hypothesis with respect to diet quality [9,12]. Study findings can thus help to understand optimal time periods for intervention to mitigate socioeconomic inequities in diet quality.

## 2. Materials and Methods 

### 2.1. Data

#### 2.1.1. Data Source and Participants

This study used cross-sectional data (2016) from the baseline wave of the Canada Food Study (CFS), a longitudinal cohort study that is examining dietary patterns and trends among youth and young adults aged 16–30 years at baseline and recruited from five Canadian urban centers: Edmonton, Halifax, Montreal, Toronto, and Vancouver [18]. Full study details are available elsewhere [18] and are briefly presented here. Trained research assistants recruited participants using an in-person intercept sampling method at local hubs (malls, transit hubs, parks, or shopping districts) in all five cities. Individuals were eligible to participate if they were between 16 and 30 years old and had Internet access through a desktop computer, laptop, or tablet. Consenting participants provided an email address to which they were sent a personalized link to a survey the next day. Respondents were discouraged from attempting to complete the survey via smartphone, although they were not restricted from doing so. No differences were observed for any primary outcomes between participants who completed the survey using a smartphone versus a computer [18]. Participants received a $2 (CAD) cash incentive at the time of recruitment and a $20 (CAD) Interac e-transfer upon survey completion. Twenty percent of individuals approached accepted the invitation to participate in the study, and of those that did accept the invitation, 48.1% completed the survey. The CFS was reviewed and approved by the University of Waterloo Research Ethics Committee (ORE# 21631) and the authors’ present use of the data was approved by the University of Calgary Health Research Ethics Board (REB17-2043_REN1). The data used for the present analysis may be obtained from the Principal Investigator of the CFS upon request (www.canadafoodstudy.ca/contact).

The survey was comprised of two parts. The first part queried participants’ sociodemographic and physiological (self-reported height and weight) characteristics. The second part consisted of a web-based 24-hour dietary recall completed using the Canadian version of the Automated Self-Administered 24-hour Dietary Assessment Tool (ASA24-Canada-2016) [18]. Following completion of the first ASA24 recall, participants were asked to complete a second recall 4–10 days later.

A total of 3000 participants took part in the first wave of the CFS. Participants who were < 18 years of age (*n* = 383) were excluded from the present analyses as the outcome of interest was adult diet quality. Participants who did not complete a 24-hour dietary recall and who were missing data on SEP in childhood or adulthood (*n* = 668) were also excluded, leaving a total of 1949 participants with complete data available for the analysis (representing 64.5% of the full CFS analytical sample; Figure 1 [18]). 

#### 2.1.2. Healthy Eating Index–2015 (HEI-2015) Scores

Participants reported dietary intake using the ASA24-Canada-2016 [19]. This online tool prompts respondents to report all foods and beverages consumed the prior day from midnight to midnight, using a modified version of the U.S. Department of Agriculture’s Automated Multiple-Pass Method. Passes include prompts to capture eating occasions and the foods and beverages consumed, along with details such as method of preparation and additions. Images are used to support estimation of portion sizes [19]. Foods and beverages reported are auto-coded using a database adapted from the Canadian Nutrient File 2015 by Health Canada [20]. ASA24-Canada output also includes values for food groups and other dietary components based on the United States Department of Agriculture’s Food Patterns Equivalents Database [21]. Together, this information allowed estimation of intakes of nutrients, food groups, and other dietary components required to calculate HEI-2015 scores [22]. 

The HEI-2015 is a measure of overall diet quality that measures alignment with the 2015–2020 Dietary Guidelines for Americans [23]. The HEI-2015 consists of nine adequacy components (including total fruits, whole fruits, total vegetables, greens and beans, whole grains, dairy, total protein foods, seafood and plant proteins, fatty acids) and four moderation components (including refined grains, sodium, added sugars, saturated fats), which are to be consumed in limited amounts relative to the total diet [17,24]. Component scores are density-based (e.g., expressed per 1000 kcal) to allow estimation of diet quality that is not confounded by energy intake. Component scores are summed to yield a total score ranging from 0 to 100, with a higher score indicating greater adherence to the Dietary Guidelines for Americans. Although a Canadian version of the HEI has been developed, it reflects the 2007 version of Eating Well with Canada’s Food Guide and is not density based [7]. Further, the HEI-2015 has undergone extensive validation and has been shown to capture the multidimensionality of diet as well as to have predictive validity [17,24]. Moreover, Canada and the United States share similar dietary cultures, and key aspects of dietary guidance have already been harmonized (e.g., the Dietary Reference Intakes, which underpin food-based dietary guidelines, are shared between the two countries).

Component and total HEI-2015 scores were calculated at the level of each individual person based on a single ASA24 recall using the publicly-available SAS (Version 9, Cary, NC, USA) macros developed by the National Cancer Institute [22]. Since not all individuals completed both ASA24 recalls, the first recall completed by each participant was used.

#### 2.1.3. Childhood and Adult SEP

Childhood SEP was assessed based on participants’ self-report of their parent(s)’/guardian(s)’ highest attained educational level by asking the following questions: “What is the highest level of formal education your mother completed?” and “What is the highest level of formal education your father completed?” Childhood SEP was denoted by the higher of the mother’s or father’s education level [25]. Parental education is a widely accepted and used indicator of childhood SEP [26]. Adult SEP was denoted by the respondent’s highest level of education currently completed or underway at the time of the survey. Current completion level was assessed by asking: “What is the highest level of formal education you have completed?”. Participants were asked if they were presently students and, if so, what level of education they were currently pursuing. For participants who were current students, we made the assumption that they would complete the educational credentials underway, and assigned SEP on the basis of those in-progress credentials [27]. For non-students, their highest level of education completed was used to denote adult SEP.

All three educational attainment questions had the same response options: less than high school diploma; high school diploma or equivalent; trade certificate or diploma from a technical/vocational school or apprenticeship training, diploma, or certificate from a community college or CEGEP (a publicly funded college system exclusive to the province of Quebec); some university or certificate/diploma below the bachelor’s level; bachelor’s degree (e.g., BA, BSc); and university degree above the bachelor’s level (e.g., master’s, professional school, doctorate). These options were collapsed into three categories for analysis: high school or less (low SEP), certificate/diploma (partial or complete) (medium SEP), and university (partial or complete) (high SEP). 

#### 2.1.4. Other Covariates

The following self-reported covariates were included in all models as potential confounders: age group (18–21, 22–25, 26–30 years), sex (female, male), race/ethnicity (White only, Chinese only, South Asian only, Black only, Aboriginal (including mixed), mixed/other/not stated/missing), and body mass index (BMI; kg/m^2^) assessed from self-reported height and weight (underweight (≤18.5 kg/m^2^), normal (>18.5–≤25.0 kg/m^2^), overweight (>25.0–≤30.0 kg/m^2^), obese (≥25.0 kg/m^2^), preferred not to respond [18]). Mode of completion of the survey (non-mobile/mobile) was also included as a covariate. Participants were instructed to use non-mobile devices to complete the survey, however, a small percentage of participants did use a mobile device. This variable was considered a potential indicator of data quality and not a potential confounder [18]. 

### 2.2. Statistical Analysis

#### 2.2.1. Sample Weighting

Data were weighted using post-stratification sample weights constructed based on population estimates for 2016 from the 2011 Canadian Census [28]. Sample probabilities were created for 30 demographic groups (age by sex) based on weighted proportions. Weights were calculated as (1/sample probability) for each group, and applied to the full dataset. Estimates reported are weighted unless otherwise specified.

Descriptive, ANOVA, regression and mediation analyses were conducted using STATA version 15 (StataCorp., College Station, TX, USA) with *p* < 0.05 indicating statistically significant results.

#### 2.2.2. Descriptive Statistics

Means and group sizes were computed for each covariate of interest. Radar graphs [24] were used to illustrate differences in overall and HEI-2015 component scores between varying levels of childhood and adult SEP. 

#### 2.2.3. Regression Analyses (to Assess Critical Periods and Cumulative Effects Hypotheses)

ANOVA tests were used to determine differences in mean HEI-2015 scores across the educational groupings. Associations between each of childhood and adult SEP and diet quality in young adulthood were assessed using regression models. First, crude bivariate linear regression models were constructed to examine unadjusted associations between SEP in childhood and adulthood with adult HEI-2015 scores (bivariate associations). University level education was used as the reference group in all models. Second, multivariable linear regression models examined associations between childhood SEP (Model 1A) and adult HEI-2015 scores and between adult SEP (Model 1B) and adult HEI-2015 scores, adjusting for age group, sex, race/ethnicity, BMI, and survey mode. Adult SEP was omitted when childhood SEP was included, and vice versa. Third, multivariable linear regression models examined associations between childhood and adult SEP with adult HEI-2015 scores, adjusting for all covariates (Model 2). 

#### 2.2.4. Mediation Analyses (to Assess Pathways Hypothesis)

We examined whether adult SEP mediated associations between childhood SEP and HEI-2015 scores in young adulthood using methods described by Iacobucci [29] for mediation analyses involving categorical exposures, mediators, and/or outcomes. The method accounts for the differing scales of linear and ordinal models by standardizing beta-coefficients using the standard deviation of the predictors and outcomes [29,30,31]. A *z*-test was applied to determine whether there was statistical evidence of mediation for each of the categories of adult SEP. 

To accompany the statistical test for presence vs. absence of mediation, subsequent mediation analyses using dummy variables for both childhood and adult SEP (high compared to medium education and high compared to low education) were also conducted to estimate the percentage of the childhood SEP and adult HEI-2015 scores relationship mediated by adult SEP. These analyses were conducted using the medeff package for Stata [32,33]. These analyses are unadjusted as the program cannot incorporate multi-categorical variables.

## 3. Results

### 3.1. Descriptive Statistics

Table 1 provides weighted and unweighted descriptive statistics for the analytic sample. Weighted results only are presented hereafter. The mean HEI-2015 score for the sample was 52.8 ± 14.8 of a total possible score of 100 points. The majority of participants had a high SEP in both childhood and adulthood, with 57.7% of participants’ parents achieving at least university-level education and 69.7% of participants attaining the same level themselves as adults. The weighted sample was most represented by 26–30-year-olds (40.3%) while 18–21 and 22–25-year-olds represented 27.8% and 22.6% of the weighted sample, respectively. Most participants identified their ethnicity as White (52.5%). The majority had a normal BMI (58.1%), while the remainder were categorized as being underweight (5.6%), overweight (19.9%), or obese (8.7%), or preferred not to respond (7.7%).

Figure 2 and Figure 3 depict HEI-2015 overall and component scores according to childhood (Figure 2) and adult SEP (Figure 3). For both childhood and adult SEP, mean HEI-2015 component scores were significantly different across SEP groups for total fruit, whole fruits, total vegetables, greens and beans, and seafood and plant proteins, with lower SEP groups having lower scores. In addition, mean scores for added sugars were significantly lower for those with a lower childhood SEP, while whole grains scores were significantly lower for those with a lower adulthood SEP. Similarly, overall HEI-2015 scores were significantly lower across declining SEP groups (*p* < 0.001).

### 3.2. Regression Results

#### 3.2.1. Bivariate Regression

Lower SEP in childhood and adulthood were both significantly associated with lower (poorer) HEI-2015 scores in young adulthood (Table 2). Compared to participants whose parents had completed University, participants whose parents had completed a certificate/diploma or high school or less had mean HEI scores that were 1.94 (95% CI: −3.65, −0.23) and 3.60 (95% CI: −5.79, −1.41) points lower, respectively, indicating poorer diet quality. Similarly, compared to participants who had attained a university-level education, participants who had completed a certificate/diploma or high school or less had mean HEI-2015 scores in adulthood that were lower by 2.45 (95% CI: −4.28, −0.62) and 6.00 (95% CI: −8.87, −3.13) points, respectively. 

#### 3.2.2. Models 1A and 1B

Models 1A and 1B extend the bivariate models by estimating the association of childhood and adult SEP with adult HEI-2015 scores adjusted for covariates (Table 3). Model 1A represents associations between childhood SEP adjusted for all covariates, and Model 1B represents associations between adult SEP adjusted for all covariates. Model 1A is not adjusted for adult SEP, and Model 1B is not adjusted for childhood SEP. In Model 1A, compared to participants whose parents had completed university, participants whose parents had completed a certificate/diploma or high school or less had mean HEI-2015 scores that were lower by 1.81 (95% CI: −3.49, −0.14) and 2.73 (95% CI: −4.96, −0.51) points, respectively. In Model 1B, compared to adults who had completed university, adults who had completed a certificate/diploma or high school or less had HEI-2015 scores that were lower by 2.11 (95% CI: −3.94, −0.28) and 4.79 (95% CI: −7.86, −1.73) points, respectively. 

#### 3.2.3. Model 2 (Critical Periods and Cumulative Effects Hypotheses)

Model 2 examined associations between childhood SEP and adult HEI-2015 score, adjusted for adult SEP and all covariates, and between adult SEP and adult HEI-2015 score, adjusted for child SEP and all covariates (Table 4), and thus estimates independent associations of childhood SEP and adult SEP with adult HEI-2015 score. Following the inclusion of adult SEP in the model, the relationship between childhood SEP and adult HEI-2015 score was attenuated, but remained statistically significant for participants whose parents had achieved an educational level of high school or less compared to university (*β*: −2.22 (95% CI: −4.41, −0.03)). When adjusted for childhood SEP, the association between adult SEP and adult HEI-2015 remained statistically significant. Compared to participants who attained university-level education, participants who attained a certificate/diploma or high school or less had mean HEI-2015 scores that were lower by 1.87 (95% CI: −3.73, −0.01) and 4.36 (95% CI: −7.38, −1.35) points, respectively.

### 3.3. Mediation Analysis (Pathways Hypothesis)

Table 4 presents results from mediation analyses. Adult SEP fulfilled the criteria for a mediator [30] as lower childhood SEP was significantly associated with lower adult SEP, and adult SEP was significantly associated with adult HEI-2015 score. Adult SEP at the level of university education mediated associations between childhood SEP and mean adult HEI-2015 score (*p* < 0.001). However, mediation was only marginally present at the adult SEP level of certificate/diploma (*p* = 0.08). Results from Model 2 (Table 2) illustrate this partial mediation as the relationship between childhood SEP and adult HEI-2015 score was only partially attenuated when adjusted for adult SEP. Up to 13.0% (1.4–13.0%) of the relationship between childhood SEP and mean adult HEI-2015 score was mediated via adult SEP.

## 4. Discussion

### 4.1. Summary of Findings

The present study adopted a life course perspective to enhance our understanding of the development of inequities in diet quality in young adulthood. In this analysis, educational attainment was used as an indicator of SEP in both childhood and adulthood. Educational attainment reflects knowledge-related assets as well as other health-related characteristics. including sense of control, problem-solving, and confidence to navigate challenges [34]. It is also a strong determinant of occupation and income, all of which shape access to material and social resources at the individual and societal levels [34,35]. 

Our results demonstrate that diet quality is poor among young adults living in Canadian cities, averaging 52.8 of a total possible score of 100 [24]. A negative gradient in diet quality was observed across declining SEP groups. Though differences across groups appear small in magnitude, from a population health perspective these differences are meaningful. The observed socioeconomic gradient in diet quality is consistent with findings in the literature. Using nationally representative data linked to the Canadian census, Garriguet et al. [7] found a similar gradient in Canadian-adapted HEI scores according to educational level, both for children and adults. Similarly, an investigation of trends in diet quality (using the American Heart Association Continuous Diet Score) over time in the United States found significant inequities in diet quality according to educational level and income [7]. Trends over time indicated that these inequities remained consistent or widened between 1999 and 2012 [7]. A substantial body of additional research demonstrates clear socioeconomic inequities in other dietary constituents such as energy intake, fruits, vegetables, fast food, and a variety of nutrients in Western nations [12,35,36]. The present findings add to existing research by investigating inequities in diet quality in young Canadian adults.

We found that both childhood and adult SEP were independently associated with adult diet quality. The lowest level of SEP in both childhood and adulthood was associated with poorer diet quality in adulthood. Moreover, adult SEP mediated up to 13.0% of the association between childhood SEP and adult diet quality, although mediation effects were only statistically significant at the lowest category of adult SEP. Study findings align with key life course perspectives and are suggestive of latent, pathway, and cumulative effects of SEP across the early life course in shaping the socioeconomic patterning of diet quality in young adulthood [3]. 

### 4.2. Alignment of Study Findings with Life Course Hypotheses 

In the current study, SEP in childhood was associated with diet quality in young adulthood, independent of adult SEP, providing support for the critical period hypothesis. Using data from an Australian longitudinal cohort of children aged 2–15 years, Gasser et al. [9] found that children’s dietary pattern trajectories represented by Australian Dietary Guidelines scores varied by childhood SEP [9]. Similarly, in a longitudinal cohort of American adolescents and young adults (14–25 years), Watts et al. [12] found that lower childhood SEP, represented by parental education level, was associated with lower fruit and vegetable and higher fast food intake in adolescence and young adulthood [12]. These findings and others have demonstrated that higher SEP in childhood is associated with higher quality dietary patterns in subsequent and adult years [37,38]. Although we set out to test the critical periods hypothesis (which posits that the effects of childhood exposures are unchanged by any subsequent events), evidence from the current study is more consistent with the sensitive period hypothesis (which posits that effects of childhood exposures are important for outcomes later in the life course but can be altered by subsequent events) [39]. Although childhood SEP was significantly associated with adult diet quality, the inclusion of adult SEP in the regression model attenuated this association. This indicates that although there are lasting impacts of exposure to low SEP in childhood on diet quality later in life, they may be partially mitigated through achievement of a higher SEP in adulthood, and more meaningfully by university-level adult educational attainment, consistent with findings from Watts et al. [12]. The current study is the first, to our knowledge, to have examined these associations with respect to diet quality.

The cumulative effects hypothesis posits that the intensity and duration of negative exposures encountered across the life course shape health and health behaviours [3]. The independent associations between low SEP at both time points across the life course provide support for the association of repeated exposures to low SEP with adult diet quality and imply that longer duration of low SEP exposure results in poorer adult diet quality [6]. With respect to intensity of exposures, our analyses revealed that the most disadvantaged social contexts, represented by educational attainment of high school or below at either time point, exerted the strongest negative impacts on diet quality in adulthood. Taken together, this evidence is interpreted as support for the cumulative effects hypothesis. Other approaches have also been used to assess the cumulative effects hypothesis [39], and the approach we adopted is consistent with recent work by Darin-Mattson et al. [6]. Consistent with their approach, we interpreted independent, statistically significant associations between SEP at each time point with adult diet quality as providing support for the cumulative effects hypothesis as this provides evidence that repeated exposure to low SEP leads to poorer dietary outcomes [6]. 

The pathways hypothesis posits that exposure to low SEP in childhood increases the likelihood of experiencing low SEP at a subsequent time point, which is itself associated with the outcome of interest [1,4,6]. We investigated the relevance of this hypothesis to socioeconomic inequities in diet quality by testing the potential mediation effect of adult SEP on the relationship between childhood SEP and adult diet quality score. Indeed, lower childhood SEP was significantly associated with the likelihood of achieving a lower SEP in adulthood, as has been evidenced by previous research [40,41], lending support to the pathways hypothesis. Moreover, adult SEP mediated up to 13.0% of the associations between childhood SEP and diet quality in young adulthood, which is also indicative of a pathway effect. Finally, adult SEP was significantly, and most strongly, associated with adult diet quality. Although no other study has conducted these analyses with diet quality as an outcome, Nettle and Bateson [41] demonstrated that about half of the association between childhood SEP and self-rated health in adulthood was due to the continuity of childhood SEP into adulthood among a longitudinal cohort of British women aged 23–42 years. Results from the present study highlight the importance of socioeconomic trajectory in the development of inequities in dietary patterns.

### 4.3. Strengths and Limitations

The use of HEI-2015 scores provides insights into how SEP shapes overall dietary quality rather than intakes of specific nutrients, foods, or other dietary components. The combination of foods and nutrients exert synergistic and possibly antagonistic effects, so using measures of overall diet quality are important to understanding how diet impacts health and therefore understanding dietary inequities. The analyses focus on diet quality in young adulthood, a period identified as one during which nutrition and health outcomes tend to become poorer [15]. In addition, diet was assessed using an online 24-hour recall tool that has been shown to capture true intakes with good accuracy among adults [19] and that has been adapted to the Canadian food supply. 

Important limitations should also be considered. The pathways hypothesis was evaluated in a separate mediation model; therefore, empirical disentanglement of all three life course hypotheses was not possible in this study. However, our aim was not to empirically disentangle these hypotheses, but rather to investigate their relevance to development of inequities in dietary patterns for the first time. Future investigations employing longitudinal data may consider methodology that considers all three hypotheses simultaneously [6,39]. The survey response rate was low; however, the sample is weighted to be representative of the Canadian urban young adult population and thus generalizability within Canada and potentially to other developed nations with similar demographic and economic characteristics, and systems of governance, is expected to be relatively high. The data are cross-sectional and as such, interpretations of causality are not possible. However, reverse causation, in which adult diet quality determined childhood or adult SEP, is unlikely. SEP is not adequately captured by any single indicator. The current study used education to represent SEP; however, findings based on other indicators may differ. Income and occupation are also important indicators of SEP, however; these indicators are not available for both childhood and adulthood in the CFS. Childhood SEP was represented by parental educational level, which is assumed to have been consistent throughout each participant’s childhood. It is possible that participants may have reported their parents’ highest lifetime educational level. However, most formal education is completed in young adulthood and parental education is a widely used and accepted indicator of childhood SEP within life course studies [26]. 

A large percentage of the sample were students, however, student status was not found to be associated with diet quality. The assumption was made that students with education in progress would complete their studies, however, it is possible that they may not complete their studies and not attain the SEP level that was assigned to them, or alternatively, that they may attain a higher level than reported at the time of the study. HEI-2015 scores were calculated based on recall data for a single day for each individual. Although a single day’s intake is unlikely to represent usual diet quality for an individual, group means, as used here, can adequately represent usual group-level intakes [42]. Methods developed to better reflect usual diet quality have been developed for the HEI, but are not yet amenable to generating individual predicted scores that can be used as dependent variables in regression modelling [17]. Additionally, dietary intake is known to be reported with error [43] and due to potential underreporting of foods considered to be less healthy and over-reporting of those considered to be more healthy, HEI-2015 scores may be overestimated. However, this would only affect the current findings if reporting differed by SEP, potentially resulting in spurious effects.

Mediation analyses using categorical exposure and mediation variables present analytical challenges that have not, to date, been fully addressed. No statistical package currently exists that can provide the percentage effect mediated using categorical exposure and mediation variables [32]. As such, for ease of interpretation, we first tested whether or not mediation was present using a method proposed by Iacobucci [29] and subsequently conducted mediation analyses using dummy variables for both the exposure and mediator to estimate the proportion mediated. 

### 4.4. Conclusions

The present results provide support for the relevance of three life course hypotheses to the development of socioeconomic inequities in diet quality in young adulthood. Evidence in support of the critical periods hypotheses demonstrates the importance of early childhood conditions in shaping diet quality across the early life course. Evidence in support of the cumulative effects and pathways hypotheses demonstrates the importance of socioeconomic conditions beyond those experienced in childhood in shaping diet quality. Although findings demonstrate the importance of life conditions experienced during both childhood and adulthood in shaping adult diet quality, adult SEP was more strongly associated with adult diet quality. Moreover, high adult SEP both partially mediated and attenuated the negative association of low childhood SEP with adult diet quality. Nevertheless, childhood SEP laid a foundation for subsequent adult SEP and diet quality. Study findings highlight the need to increase access to post-secondary education for young adults, and particularly for those of low SEP backgrounds [12,15]. Future research may consider assessing the relevance of life course hypotheses using longitudinal data and methodology that allows the opportunity to simultaneously consider all three life course hypotheses to better elucidate the relationship between SEP and diet quality across the life course [39]. 

## Figures and Tables

**Figure 1 nutrients-11-00242-f001:**
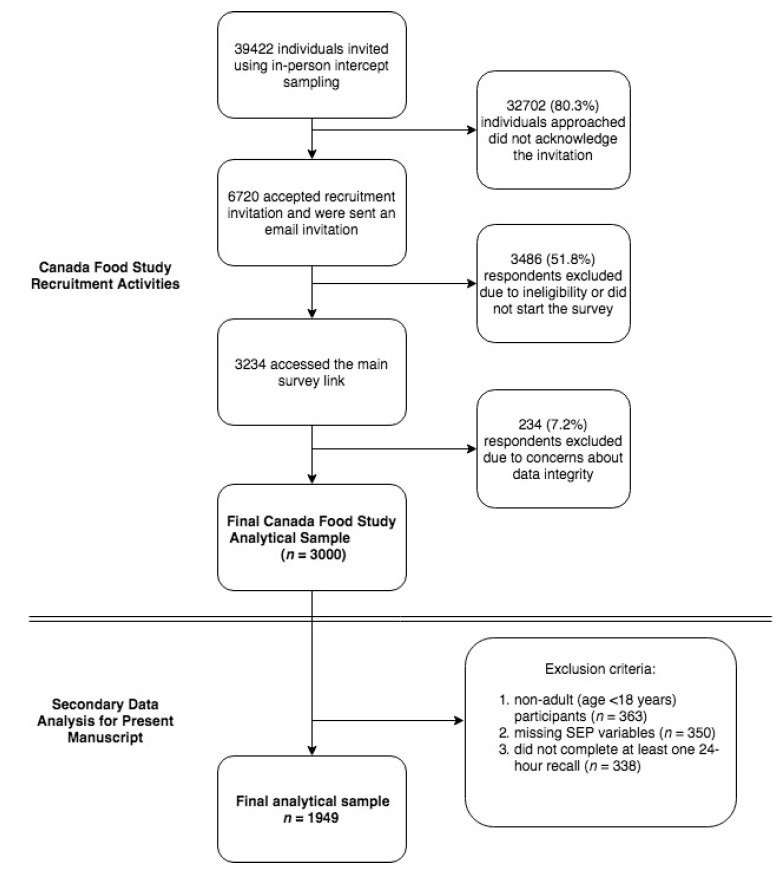
Flowchart describing Canada Food Study (CFS) sample selection and sample selection for the present analysis. SEP: socioeconomic position.

**Figure 2 nutrients-11-00242-f002:**
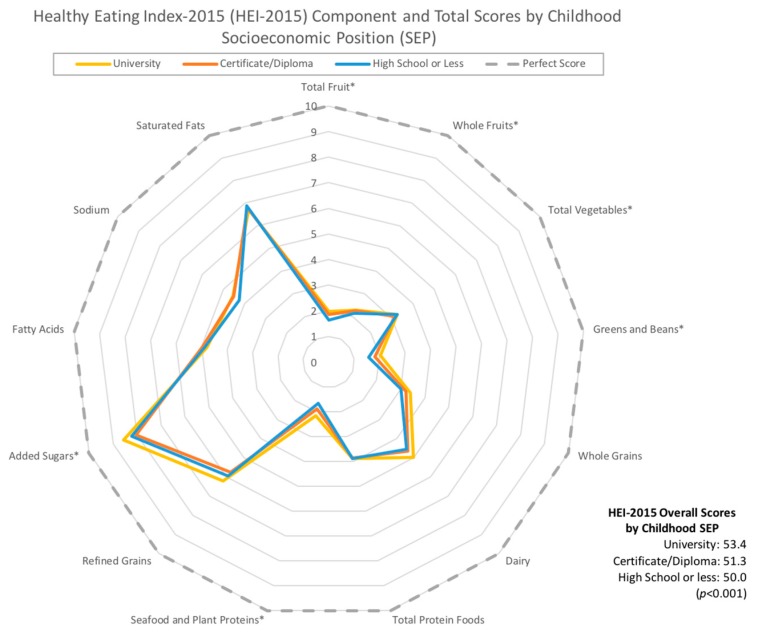
Radar graph depicting component scores across educational levels of childhood socioeconomic position (SEP). Asterisks (*) beside component names indicate that the differences in mean Healthy Eating Index–2015 (HEI-2015) component scores across groups were statistically significant using an analysis of variance (ANOVA) model.

**Figure 3 nutrients-11-00242-f003:**
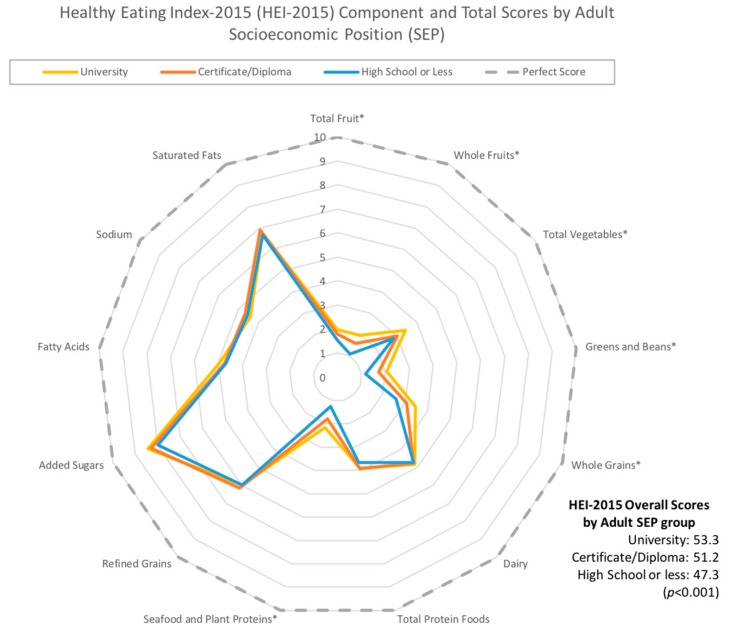
Radar graph depicting component scores across educational levels of adult socioeconomic position (SEP). Asterisks (*) beside component names indicate that the differences in mean Healthy Eating Index–2015 (HEI-2015) component scores across groups were statistically significant using an analysis of variance (ANOVA) model.

**Table 1 nutrients-11-00242-t001:** Participants in the 2016 wave of the Canada Food Study (*n* = 1949).

Variable	Unweighted	Weighted
	**Mean (Standard Deviation (SD))**	**Mean (SD)**
**Healthy Eating Index–2015 (HEI-2015)**	52.3 (15.0)	52.8 (14.8)
**Parent/Guardian Education Level (Childhood Socioeconomic Position (SEP))**	**% (*n*)**	**%**
University (partial or complete)	57.7 (1124)	56.5
Certificate/diploma (partial or complete)	25.9 (506)	26.5
High school or less	16.4 (319)	17.0
**Current Education Level (Adult SEP)**		
University (partial or complete)	68.8 (1341)	69.7
Certificate/diploma (partial or complete)	22.2 (433)	22.0
High school or less	9.0 (175)	8.2
**Age**		
18–21 years	45.7 (891)	27.8
22–25 years	31.7 (617)	31.9
26–30 years	22.6 (441)	40.3
**Current Student**		
No	33.8 (658)	45.0
Yes	66.2 (1291)	55.0
**Sex**		
Female	62.5 (1219)	51.2
Male	37.5 (730)	48.8
**Ethnicity**		
White only	51.8 (1010)	52.5
Chinese only	8.7 (169)	8.6
South Asian only	7.1 (138)	7.6
Black only	4.8 (94)	4.2
Aboriginal (including mixed)	3.5 (68)	3.2
Mixed/other/not stated	24.1 (470)	23.8
**Body Mass Index (BMI) Group**		
Normal	57.6 (1122)	58.1
Underweight	6.8 (133)	5.6
Overweight	18.5 (360)	19.9
Obese	8.7 (170)	8.7
Preferred not to respond	8.4 (164)	7.7
**Survey Mode**		
Mobile	12.3 (240)	11.4
Non-mobile	87.7 (1709)	88.6

**Table 2 nutrients-11-00242-t002:** Bivariate (unadjusted) associations between all variables and Healthy Eating Index-2015 (HEI-2015) scores in young adulthood.

	Bivariate (Unadjusted)	
**Parameter**	**β (95% CI)**	***p*-value**
**Parent/Guardian Education Level (Childhood Socioeconomic Position (SEP))**		
University (ref)	-	-
Certificate/diploma	**−1.94 (−3.65, −0.23)**	**0.03**
High school or less	**−3.60 (−5.79, −1.41)**	**0.001**
**Current Education Level (Adult SEP)**		
University (ref)	-	-
Certificate/diploma	**−2.45 (−4.28, −0.62)**	**0.009**
High school or less	**−6.00 (−8.87, −3.13)**	**<0.001**
**Age**		
18–21 years (ref)	-	-
22–25 years	**1.67 (0.08, 3.26)**	**0.04**
26–30 years	**3.76 (2.01, 5.51)**	**<0.001**
**Current Student**		
No (ref)	-	-
Yes	−0.19 (−1.72, 1.34)	0.81
**Sex**		
Female (ref)	-	-
Male	**−2.24 (−3.73, −0.75)**	**0.003**
**Race**		
White only (ref)	-	-
Chinese only	**−5.30 (−7.68, −2.92)**	**<0.001**
South Asian only	**−4.58 (−7.44, 1.72)**	**0.002**
Black only	−2.58 (−6.25, 1.09)	0.17
Aboriginal (including mixed)	**−4.66 (−8.26, −1.07)**	**0.01**
Mixed/other/not stated/missing	**−2.60 (−4.52, −0.68)**	**0.008**
**Body Mass Index (BMI) Group**		
Normal (ref)	-	-
Underweight	**−2.95 (−5.67, −0.23)**	**0.03**
Overweight	−1.63 (−3.63, 0.36)	0.11
Obese	**−4.02 (−6.38, −1.65)**	**0.001**
Preferred not to respond	**−3.31 (−6.03, −0.59)**	**0.02**
**Mobile User, %**		
No (ref)	-	-
Yes	**−2.55 (−5.04, −0.07)**	**0.04**

Results in bold are statistically significant where *p* < 0.05. CI: Confidence Interval.

**Table 3 nutrients-11-00242-t003:** Multivariable associations (adjusted) between socioeconomic position (SEP) in childhood and young adulthood with Healthy Eating Index-2015 (HEI-2015) scores in young adulthood.

	Model 1A		Model 1B		Model 2	
**Parameter**	***β* (95% CI)**	***p*-value**	***β* (95% CI)**	***p*-value**	***β* (95% CI)**	***p*-value**
**Parent/Guardian Education Level (Childhood SEP)**						
University (ref)	-	**-**	-	-	**-**	**-**
Certificate/diploma	**−1.81 (−3.49, −0.14)**	**0.03**	-	-	−1.57 (−3.27, −0.13)	0.07
High school or less	**−2.73 (−4.96, −0.51)**	**0.02**	-	-	**−2.22 (−4.41, −0.03)**	**0.047**
**Current Education Level (Adult SEP)**						
University (ref)	-	-	-	-	-	-
Certificate/diploma	-	-	**−2.11 (−3.94, −0.28)**	**0.02**	**−1.87 (−3.73, −0.01)**	**0.048**
High school or less	-	-	**−4.79 (−7.86, −1.73)**	**0.002**	**−4.36 (−7.38, −1.35)**	**0.005**
**Age**						
18–21 years (ref)	-	-	-	-	-	-
22–25 years	**1.87 (0.21, 3.52)**	**0.03**	1.50 (−0.18, 3.19)	0.08	1.53 (−0.16, 3.21)	0.08
26–30 years	**4.45 (2.50, 6.40)**	**<0.001**	**3.85 (1.80, 5.91)**	**<0.001**	**3.92 (1.88, 5.97)**	**<0.001**
**Current Student**						
No (ref)	-	-	-	-	-	-
Yes	**1.72 (0.02, 3.43)**	**0.047**	1.31 (−0.47, 3.09)	0.15	1.22 (−0.57, 3.01)	0.18
**Sex**						
Female (ref)	-	-	-	-	-	-
Male	**−1.82 (−3.30, −0.04)**	**0.02**	**−1.69 (−3.17, −0.21)**	**0.03**	**−1.64 (−3.12, −0.17)**	**0.03**
**Race**						
White only (ref)	-	-	-	-	-	-
Chinese only	**−4.91 (−7.31, −2.50)**	**<0.001**	**−5.53 (7.88, −3.19)**	**<0.001**	**−5.32 (−7.71, −2.93)**	**<0.001**
South Asian only	**−4.24 (−7.08, −1.40)**	**0.003**	**−4.58 (7.44, 1.71)**	**0.002**	**−4.42 (−7.26, −1.59)**	**0.002**
Black only	−1.60 (−5.35, 2.15)	0.40	−1.68 (−5.49, 2.13)	0.39	−1.51 (−5.38, 2.37)	0.45
Aboriginal (incl. mixed)	−2.83 (−6.61, 0.95)	0.14	−2.72 (−6.51, 1.05)	0.16	−2.28 (−6.08, 1.52)	0.24
Mixed/other/not stated/missing	**−2.33 (−4.27, −0.39)**	**0.02**	**−2.16 (4.10, −0.22)**	**0.03**	**−2.23 (−4.16, −0.29)**	**0.02**
**Body Mass Index (BMI) Group**						
Normal (ref)	-	-	-	-	-	-
Underweight	−2.08 (−4.81, 0.64)	0.13	−2.02 (−4.80, 0.76)	0.15	−1.96 (−4.71, 0.80)	0.16
Overweight	−0.95 (−2.91, 1.02)	0.35	−1.08 (−3.05, 0.88)	0.28	−0.87 (−2.83, 1.09)	0.38
Obese	**−4.03 (−6.35, −1.71)**	**0.001**	**−4.03 (−6.36, −1.70)**	**0.001**	**−3.97 (−6.31, −1.64)**	**0.001**
Preferred not to respond	−2.70 (−5.42, 0.03)	0.053	−2.66 (−5.40, 0.09)	0.06	−2.42 (−5.16, 0.33)	0.09
**Mobile User, %**						
No (ref)	-	-	-	-	-	-
Yes	−1.92 (−4.32, 0.48)	0.12	−1.32 (−3.70, 1.05)	0.28	−1.27 (−3.63, 1.09)	0.29

Model 1A is a multivariate model representing associations between childhood SEP adjusted for all covariates, with the exception of adult SEP. Model 1B is a multivariate model representing associations between adult SEP adjusted for all covariates, with the exception of childhood SEP. Model 2 represents a fully adjusted model. Results in bold are statistically significant, where *p* < 0.05. CI: confidence interval.

**Table 4 nutrients-11-00242-t004:** Tests for the potential mediation of the association between childhood socioeconomic position (SEP) and mean Healthy Eating Index-2015 (HEI-2015) score in young adulthood by SEP in young adulthood.

Association of Childhood SEP with Adult SEP	OR (95% CI)		
University (ref)	-		
Certificate/diploma	**1.75 (1.32, 2.32)**		
High school or less	**2.40 (1.71, 3.37)**		
**Association of Adult SEP with Mean Adult HEI-2015 Score Adjusted for Childhood SEP**	**β (95% CI)**		
University (ref)	-		
Certificate/diploma	**−1.87 (−3.73, −0.13)**		
High school or less	**−4.36 (−7.38, −1.35)**		
**Test for Mediation by Adult SEP of the Association between Childhood SEP and Adult HEI-2015 Score**	**Standardized Standard Error**	**Z-statistic**	***p*-value**
University (ref)	-	-	-
Certificate/diploma	4.73	−1.75	0.08
High school or less	5.46	**−2.44**	**<0.001**

All models adjusted for age, student status, sex, race/ethnicity, body weight status, and survey mode. Results in bold are statistically significant where *p* < 0.05. OR: odds ratio, CI: confidence interval.

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
