# Peer review of "Socioeconomic Disadvantage across the Life Course Is Associated with Diet Quality in Young Adulthood"

_nutrients, 2019, doi:10.3390/nu11020242_

Round 1
Reviewer 1 Report
The purpose of this study was to investigate the associations between SEP in both childhood and young adulthood with diet quality in young adulthood. The authors have prepared a well-written manuscript on an important issue, but there are some areas that require more thought before this manuscript is ready for publication.
Major comments:
1) In this manuscript the authors are investigating three-different life course hypotheses (e.g. pathway, cumulative and critical periods) in relation to their research question. While it is good to see that the authors are testing multiple life course models in the one study, I have some concerns with the statistical approach they have taken to test these life course hypotheses, as well as some of the conclusions they are inferring from their results. In the discussion section, the authors discuss the impact of longer duration of SEP on diet quality. In my opinion, you can’t state this as you didn’t sum the time periods (childhood and adult SEP) in your analysis. Further to this (on page 12, lines 371-374), the authors state, “This indicates that although there are lasting impacts of exposure to low SEP in childhood on dietary patterns later in life, they may be partially mitigated through achievement of a higher SEP in adulthood, and more meaningfully by University level adult educational attainment”. I think the authors need to be careful stating this because they didn’t test this. The authors would need to group individuals based on the SEP mobility (social mobility) from childhood to adulthood and then test these hypotheses with them grouped. So, the authors either need to re-write some of these statements or consider the approach I have suggested below.
The following paper (see Mishra et al 2008 below) describes a model selection approach to delineate the different life course hypotheses. This approach is becoming more common in the current life course literature and the authors should consider using this approach for their analyses (or something along these lines) to disentangle the life course hypotheses in relation to their research questions. This would also set their work apart from what has already been published to date in this area and will make the justification and need for this manuscript much stronger.
Mishra, G., Nitsch, D., Black, S., De Stavola, B., Kuh, D. and Hardy, R., 2008. A structured approach to modelling the effects of binary exposure variables over the life course. International journal of epidemiology, 38(2), pp.528-537.
On page 2, line 56, the authors stated, “that cannot necessarily be empirically disentangled”. Using the approach described in Mishra’s paper (above) will allow the authors to disentangle and unpack the life course models described in their manuscript better and will allow for stronger conclusions.
Also, if the authors were to consider the above approach they may not have to do both the regression and mediation analyses. This is because the structured modelling approach considers all models (life course hypotheses) in the same analyses.
2) The authors need to revisit the definition/explanation of critical periods and assess whether this is the most appropriate hypothesis for what they are describing in their manuscript. In my opinion, I think a sensitive period would be more appropriate in this case. See Ben-Shlomo Y, Kuh D. A life course approach to chronic disease epidemiology: conceptual models, empirical challenges, and interdisciplinary perspectives. Int J Epidemiol. 2002;31:285–93. If the authors choose to change to a sensitive period, please replace the words critical period throughout the manuscript.
Could the authors also clarify what they mean by the “pathways hypothesis” in the introduction. Is this also referred to as the social mobility hypothesis? Some clarification is needed.
3) In the introduction, the authors state “To our knowledge, this is the first study to examine…..”, however, references 8, 11 and 12 looks as though they have investigated a similar question. What is your point of difference? Be more specific here.
Minor comments
1) Page 3, line 109 and in Table 1, the authors use the words baseline data. I would consider removing this or changing this word as it makes it seem like you are using longitudinal data in this manuscript. This could be confusing for the reader (given that you are exploring life course hypothesis, the reader might assume this is longitudinal data).
2) Consider adding a flow chart in your methods/results section to describe the sample selection.
3) Page 4, lines 170 and 171, the authors state “Childhood SEP was denoted by the higher of the mothers or fathers education level.” Do you have any literature to support this method of doing this?
4) Briefly state how you categorised the BMI categories (based on what cut-offs)?
5) The authors have included mode of completion of the survey as a potential confounder. Does this variable fit the criteria for a confounder?
6) The authors should consider removing the text on page 10, lines 291 (starting at “The”)-294. This information is in the table and doesn’t need to be in the text as well.
7) The authors beta coefficients on lines 297 and 298 are different to what is in Table 2. The authors need to check these and correct them where necessary.
8) The last paragraph on page 10 (lines 299-305) needs some context. Why are you describing these results after you have already spoken about Model 2 in relation to your research questions?
9) Change the word “intensity” on page 12, line 352. This word doesn’t make sense in this context and is confusing.
Author Response
Comments and Suggestions for Authors
The purpose of this study was to investigate the associations between SEP in both childhood and young adulthood with diet quality in young adulthood. The authors have prepared a well-written manuscript on an important issue, but there are some areas that require more thought before this manuscript is ready for publication.
Sincere thanks for your thoughtful feedback on this manuscript. Our responses are provided in red text below your comments.
Major comments:
1) In this manuscript the authors are investigating three-different life course hypotheses (e.g. pathway, cumulative and critical periods) in relation to their research question. While it is good to see that the authors are testing multiple life course models in the one study, I have some concerns with the statistical approach they have taken to test these life course hypotheses, as well as some of the conclusions they are inferring from their results. In the discussion section, the authors discuss the impact of longer duration of SEP on diet quality. In my opinion, you can’t state this as you didn’t sum the time periods (childhood and adult SEP) in your analysis. Further to this (on page 12, lines 371-374), the authors state, “This indicates that although there are lasting impacts of exposure to low SEP in childhood on dietary patterns later in life, they may be partially mitigated through achievement of a higher SEP in adulthood, and more meaningfully by University level adult educational attainment”. I think the authors need to be careful stating this because they didn’t test this. The authors would need to group individuals based on the SEP mobility (social mobility) from childhood to adulthood and then test these hypotheses with them grouped. So, the authors either need to re-write some of these statements or consider the approach I have suggested below.
The following paper (see Mishra et al 2008 below) describes a model selection approach to delineate the different life course hypotheses. This approach is becoming more common in the current life course literature and the authors should consider using this approach for their analyses (or something along these lines) to disentangle the life course hypotheses in relation to their research questions. This would also set their work apart from what has already been published to date in this area and will make the justification and need for this manuscript much stronger.
Mishra, G., Nitsch, D., Black, S., De Stavola, B., Kuh, D. and Hardy, R., 2008. A structured approach to modelling the effects of binary exposure variables over the life course. International journal of epidemiology, 38(2), pp.528-537.
On page 2, line 56, the authors stated, “that cannot necessarily be empirically disentangled”. Using the approach described in Mishra’s paper (above) will allow the authors to disentangle and unpack the life course models described in their manuscript better and will allow for stronger conclusions.
Also, if the authors were to consider the above approach they may not have to do both the regression and mediation analyses. This is because the structured modelling approach considers all models (life course hypotheses) in the same analyses.
We sincerely appreciate this assessment and the sharing of further literature on this topic. Please see our responses to your points below:
1) With respect to the cumulative effects hypothesis, we have included support from a recent study (Darin-Mattson et al, 2018, reference 6) who, in investigating the relevance of life course hypotheses for psychological distress throughout the life course, state: “Statistically significant, independent associations between each measured point of financial hardship and psychological distress in old age would be interpreted as support for accumulation of risks (i.e. cumulative effects) hypothesis.” In reference to this work, we also interpret independent, statistically significant associations between SEP in childhood and adult diet quality as being in support of the cumulative effects hypothesis as this would provide evidence that repeated exposures to low SEP across the life course results in poorer dietary outcomes. However, we do acknowledge that the methodology employed by Mishra as pointed out and by Darin-Mattson allow all three hypotheses to be simultaneously tested, while ours assesses the relevance of the pathways hypothesis separately. However, our aim was not to empirically disentangle these hypotheses, but rather to investigate their relevance to inequities in dietary patterns for the first time in the literature. We acknowledge the limitation of our approach to this analysis in the limitations section (lines 965-970) and encourage future analyses that build on the present work to consider longitudinal data and the methodology that you suggest to empirically disentangle the relevance of these three hypotheses for the development of dietary inequities.
Changes with respect to this comment:
Lines 121-126: We also hypothesized that childhood and young adult SEP would each be independently associated with diet quality in young adulthood which we interpreted as providing evidence in support of the cumulative effects hypothesis as this suggests that repeated exposure to low SEP leads to poor dietary outcomes [6], and that the effects of SEP would be most pronounced for those with the lowest educational attainment (suggesting greater intensity of negative exposures).
Lines 810-823: The cumulative effects hypothesis posits that the intensity and duration of negative exposures encountered across the life course shape health and health behaviours [3]. The independent associations between low SEP at both time points across the life course provide support for the association of repeated exposures to low SEP with adult diet quality and imply that longer duration of low SEP exposure results in poorer adult diet quality [6]. With respect to intensity of exposures, our analyses revealed that the most disadvantaged social contexts, represented by educational attainment of High School or below at either time point, exerted the strongest negative impacts on diet quality in adulthood. Taken together, this evidence is interpreted as support for the cumulative effects hypothesis. Other approaches have also been used to assess the cumulative effects hypothesis [39], and the approach we adopted is consistent with recent work by Darin-Mattson et al [6]. Consistent with their approach, we interpreted independent, statistically significant associations between SEP at each time point with adult diet quality as providing support for the cumulative effects hypothesis as this provides evidence that repeated exposure to low SEP leads to poorer dietary outcomes [6].
Lines 965-970: The pathways hypothesis was evaluated in a separate mediation model; therefore, empirical disentanglement of all three life course hypotheses was not possible in this study. However, our aim was not to empirically disentangle these hypotheses, but rather to investigate their relevance to development of inequities in dietary patterns for the first time. Future investigations employing longitudinal data may consider methodology that considers all three hypotheses simultaneously [6,39]
2) With respect to your comments about the mitigation of the negative effects of a low childhood SEP by higher educational attainment in adulthood, we more clearly explain how we concluded this based on the analyses we conducted and cite literature that supports our findings.
Lines 803-809: Although childhood SEP was significantly associated with adult diet quality, the inclusion of adult SEP in the regression model attenuated this association. This indicates that although there are lasting impacts of exposure to low SEP in childhood on dietary patterns later in life, they may be partially mitigated through achievement of a higher SEP in adulthood, and more meaningfully by University level adult educational attainment, consistent with findings from Watts et al [12]. The current study is the first, to our knowledge, to have examined these associations with respect to diet quality.
3) We have also removed the statement about the hypotheses not being able to necessarily be empirically disentangled and instead emphasized the key purpose of the sentence which is to indicate that the hypotheses are not mutually exclusive (line 55-56).
2) The authors need to revisit the definition/explanation of critical periods and assess whether this is the most appropriate hypothesis for what they are describing in their manuscript. In my opinion, I think a sensitive period would be more appropriate in this case. See Ben-Shlomo Y, Kuh D. A life course approach to chronic disease epidemiology: conceptual models, empirical challenges, and interdisciplinary perspectives. Int J Epidemiol. 2002;31:285–93. If the authors choose to change to a sensitive period, please replace the words critical period throughout the manuscript.
Thank you for this point. Our research question aimed to investigate the critical periods hypothesis, but our results seem more supportive of the sensitive periods hypothesis. Given this, we have added additional interpretation in the discussion to this point:
Lines 798-809: Although we set out to test the critical periods hypothesis (which posits that the effects of childhood exposures are unchanged by any subsequent events), evidence from the current study is more consistent with the sensitive period hypothesis (which posits that effects of childhood exposures are important for outcomes later in the life course but can be altered by subsequent events) [39]. Although childhood SEP was significantly associated with adult diet quality, the inclusion of adult SEP in the regression model attenuated this association. This indicates that although there are lasting impacts of exposure to low SEP in childhood on diet quality later in life, they may be partially mitigated through achievement of a higher SEP in adulthood, and more meaningfully by University level adult educational attainment, consistent with findings from Watts et al [12]. The current study is the first, to our knowledge, to have examined these associations with respect to diet quality.
Could the authors also clarify what they mean by the “pathways hypothesis” in the introduction. Is this also referred to as the social mobility hypothesis? Some clarification is needed.
Thank you for this point. Though the concept of social mobility and the pathways hypotheses are complementary, they also have distinct features. We have added some clarifying information about the pathways hypothesis:
Lines 60-63: The pathways hypothesis posits that the effect of disadvantage is indirect by setting individuals on adverse social trajectories that continue across the life course, where one negative factor increases the possibility of experiencing the same negative factor at a subsequent time point (ie: low childhood SEP increases the possibility of low adult SEP) [2,6].
This hypothesis is complementary but distinct from the concept of social mobility which refers to the consequences of transitioning from one SEP to another, independent of the effects of SEP at either time point (van der Waal, J.; Daenekindt, S.; de Koster, W. Statistical challenges in modelling the health consequences of social mobility: the need for diagonal reference models. Int J Public Health 2017, 62, 1029-1037, doi:10.1007/s00038-017-1018-x.). However, we made the decision not to introduce discussion of social mobility in the introduction as it is not a part of the research question addressed by our manuscript and may cause confusion for the reader.
3) In the introduction, the authors state “To our knowledge, this is the first study to examine…..”, however, references 8, 11 and 12 looks as though they have investigated a similar question. What is your point of difference? Be more specific here.
Thank you for this point. The novelty of our study is to examine the hypothesis in question with respect to a diet quality index (HEI-2015), previous studies have only examined this question with respect to individual foods or diet-related behaviours, or investigated the relevance of only one life course hypothesis. Please see the change made below for clarification:
Lines 128-133: .To our knowledge, this is the first study to examine the socioeconomic patterning of diet quality specifically with respect to these three life course hypotheses. Previous work that has investigated the relevance of these hypotheses to dietary inequities have focused on specific nutrients, foods, or, diet-related behaviours (e.g., breakfast consumption) as outcomes, or has only addressed the relevance of one life course hypothesis with respect to diet quality [9,12].
Minor comments
1) Page 3, line 109 and in Table 1, the authors use the words baseline data. I would consider removing this or changing this word as it makes it seem like you are using longitudinal data in this manuscript. This could be confusing for the reader (given that you are exploring life course hypothesis, the reader might assume this is longitudinal data).
Thank you for this point, we agree that this may be confusing. Given this we have changed the sentence in question to read: This study used cross-sectional data (2016-17) from the baseline wave of the Canada Food Study (CFS), a longitudinal cohort study that is examining dietary patterns and trends among youth and young adults aged 16-30 years at baseline and recruited from five Canadian urban centres: Edmonton, Halifax, Montreal, Toronto, and Vancouver [18]. (lines 139-142).
We have also edited the title of Table 1 to be: Participants in the 2016 wave of the Canada Food Study (n=1,949).
2) Consider adding a flow chart in your methods/results section to describe the sample selection.
We have now done so, thank you for the suggestion. Please see Figure 1.
3) Page 4, lines 170 and 171, the authors state “Childhood SEP was denoted by the higher of the mothers or fathers education level.” Do you have any literature to support this method of doing this?
Thank you for this. We have added more information to this section (lines 238-242). The aim was to create an indicator for highest level of parental education in the household. Parental education is considered to be a widely accepted and valid indicator of childhood SEP (reference 26). It is generally characterized according to the highest level of educational attainment in the household, as this is the most conservative approach (reference 25).
4) Briefly state how you categorised the BMI categories (based on what cut-offs)?
We have now updated this sentence to describe the cutoffs for BMI categories: underweight [≤18.5 kg/m2], normal [>18.5 – ≤25.0 kg/m2], overweight [>25.0 - ≤30.0 kg/m2], obese [≥25.0 kg/m2] (from the CFS Technical Report, reference 18).
5) The authors have included mode of completion of the survey as a potential confounder. Does this variable fit the criteria for a confounder?
Indeed it does not meet the criteria for a confounder. Lines 148-151 and 266-268 describe that participants were instructed to use non-mobile devices to complete the survey as the survey tool was not optimized for mobile devices, but a small percentage of participants did potentially use a mobile device to complete the survey as they were not restricted from doing so. We have further explained the necessity of including this variable in the analysis as it is a potential indicator of data quality and clarified that it is not intended to act as a confounder (lines 266-268). We have also included a statement to indicate that major outcomes did not differ across survey modes in the CFS (lines 148-151).
6) The authors should consider removing the text on page 10, lines 291 (starting at “The”)-294. This information is in the table and doesn’t need to be in the text as well.
We have removed the text as suggested, as well as the subsequent sentences until line 298 as they express the same information. We have added relevant text where needed (lines 397-401).
7) The authors beta coefficients on lines 297 and 298 are different to what is in Table 2. The authors need to check these and correct them where necessary.
Thank you for pointing out this error. We have removed the text in those lines but have also thoroughly compared the information in the tables with what appears in the text to ensure harmony.
8) The last paragraph on page 10 (lines 299-305) needs some context. Why are you describing these results after you have already spoken about Model 2 in relation to your research questions?
We appreciate this point. As this information is not relevant to the research questions we have removed this paragraph.
9) Change the word “intensity” on page 12, line 352. This word doesn’t make sense in this context and is confusing.
We can see how this word is confusing in this context. This sentence has been altered to read: The lowest level of SEP in both childhood and adulthood was associated with poorer diet quality in adulthood (line 783).

Reviewer 2 Report
The authors aims to test three life-course theories: critical period, cumulative effects, and pathway using data from the Canada Food Study. The questions investigated in this paper are interesting and important, yet there are several important issues that need to be addressed.
1. Most importantly, the analyses are not exactly in line with the hypotheses supposed to test. The author(s) claimed that they tested the critical period hypothesis by looking at the relationship between childhood SEP and young adulthood diet quality. Yet a significant association between the two variables alone (i.e., results in Model 1) is insufficient to support the critical period hypothesis which predicts that adverse exposures during particular developmental periods influence later-in-life outcomes independent of subsequent events. Evidence to support this hypothesis is actually in Model 2, which shows that the childhood SEP still significantly predicts the outcome after controlling for young adulthood SEP. This needs to be clarified in the revision.
Moreover, showing that both childhood SEP and young adulthood SEP predict young adulthood quality is not sufficient to support the cumulative effects hypothesis which predicts that duration and intensity of adverse exposures matter for subsequent health outcomes. See the following studies for testing this hypothesis:
Hallqvist, Johan, John Lynch, Mel Bartley, Thierry Lang, and David Blane. 2004. “Can We Disentangle Life Course Processes of Accumulation, Critical Period and Social Mobility? An Analysis of Disadvantaged Socio-Economic Positions and Myocardial Infarction in the Stockholm Heart Epidemiology Program.” Social Science and Medicine 58(8):1555–62.
Heraclides, A., and E. Brunner. 2010. “Social Mobility and Social Accumulation across the Life Course in Relation to Adult Overweight and Obesity: The Whitehall II Study.” Journal of Epidemiology and Community Health 64(8):714–19.
Liu, Hexuan and Guang Guo. 2015. “Lifetime Socioeconomic Status, Historical Context, and Genetic Inheritance in Shaping Body Mass in Middle and Late Adulthood.” American Sociological Review 80(4):705-737.
2. While I agree that education is an important indicator of SEP, it is insufficient to draw conclusion on the relationship between SEP and diet quality merely based on education. At the minimum, analyses based on other SEP measures such as occupation and income are needed to check the robustness of the results.
3. Results in Table 2 are very confusing. What are the coefficients for the covariates in the first column of the table label as “Univariates”? Are these bivariate correlations between each covariate and the outcome variable? If so, I suggest use a different table to show the bivariate relationships. Presenting bivariate results and multivariate results in the same table is confusing. Also, results for two models (i.e., one testing for the relationship between childhood SEP and diet quality and the other testing for the relationship between young adulthood SEP and diet quality) are presented together in the column labelled as “Model 1.” It is unclear what the betas for covariates in this column stand for? I suggest present the results in two separate columns.
4. Limitations on the low response rate (20% responded and 48.1% of the responded completed the survey) need to be discussed.
Author Response
Comments and Suggestions for Authors
The authors aims to test three life-course theories: critical period, cumulative effects, and pathway using data from the Canada Food Study. The questions investigated in this paper are interesting and important, yet there are several important issues that need to be addressed.
Sincere thanks for your thoughtful feedback on this manuscript.
1. Most importantly, the analyses are not exactly in line with the hypotheses supposed to test. The author(s) claimed that they tested the critical period hypothesis by looking at the relationship between childhood SEP and young adulthood diet quality. Yet a significant association between the two variables alone (i.e., results in Model 1) is insufficient to support the critical period hypothesis which predicts that adverse exposures during particular developmental periods influence later-in-life outcomes independent of subsequent events. Evidence to support this hypothesis is actually in Model 2, which shows that the childhood SEP still significantly predicts the outcome after controlling for young adulthood SEP. This needs to be clarified in the revision.
Moreover, showing that both childhood SEP and young adulthood SEP predict young adulthood quality is not sufficient to support the cumulative effects hypothesis which predicts that duration and intensity of adverse exposures matter for subsequent health outcomes. See the following studies for testing this hypothesis:
Hallqvist, Johan, John Lynch, Mel Bartley, Thierry Lang, and David Blane. 2004. “Can We Disentangle Life Course Processes of Accumulation, Critical Period and Social Mobility? An Analysis of Disadvantaged Socio-Economic Positions and Myocardial Infarction in the Stockholm Heart Epidemiology Program.” Social Science and Medicine 58(8):1555–62.
Heraclides, A., and E. Brunner. 2010. “Social Mobility and Social Accumulation across the Life Course in Relation to Adult Overweight and Obesity: The Whitehall II Study.” Journal of Epidemiology and Community Health 64(8):714–19.
Liu, Hexuan and Guang Guo. 2015. “Lifetime Socioeconomic Status, Historical Context, and Genetic Inheritance in Shaping Body Mass in Middle and Late Adulthood.” American Sociological Review 80(4):705-737.
We sincerely appreciate these thoughts with respect to the research question and have carefully considered this, as well as reviewed the suggested literature and included it in our revisions. We have made the following changes:
1) With respect to conclusions about the critical periods hypothesis, we have reframed our conclusions to be related to the independent associations of both childhood and adult SEP rather than the Model 1 results that are not mutually adjusted. (lines 790-1: In the current study, SEP in childhood was associated with diet quality in young adulthood, independent of adult SEP, providing support for the critical periods hypothesis.)
2) With respect to the cumulative effects hypothesis, we have included support from a recent study (Darin-Mattson et al, 2018, reference 6) who, in investigating the relevance of life course hypotheses for psychological distress throughout the life course, state: “Statistically significant, independent associations between each measured point of financial hardship and psychological distress in old age would be interpreted as support for accumulation of risks (i.e. cumulative effects) hypothesis.” In reference to this work, we also interpret independent, statistically significant associations between SEP in childhood and adult diet quality as being in support of the cumulative effects hypothesis as this would provide evidence that repeated exposures to low SEP across the life course results in poorer dietary outcomes.
Changes with respect to this comment:
Lines 121-126: We also hypothesized that childhood and young adult SEP would each be independently associated with diet quality in young adulthood which we interpreted as providing evidence in support of the cumulative effects hypothesis as this suggests that repeated exposure to low SEP leads to poor dietary outcomes [6], and that the effects of SEP would be most pronounced for those with the lowest educational attainment (suggesting greater intensity of negative exposures).
Lines 810-823: The cumulative effects hypothesis posits that the intensity and duration of negative exposures encountered across the life course shape health and health behaviours [3]. The independent associations between low SEP at both time points across the life course provide support for the association of repeated exposures to low SEP with adult diet quality and imply that longer duration of low SEP exposure results in poorer adult diet quality [6]. With respect to intensity of exposures, our analyses revealed that the most disadvantaged social contexts, represented by educational attainment of High School or below at either time point, exerted the strongest negative impacts on diet quality in adulthood. Taken together, this evidence is interpreted as support for the cumulative effects hypothesis. Other approaches have also been used to assess the cumulative effects hypothesis [39], and the approach we adopted is consistent with recent work by Darin-Mattson et al [6]. Consistent with their approach, we interpreted independent, statistically significant associations between SEP at each time point with adult diet quality as providing support for the cumulative effects hypothesis as this provides evidence that repeated exposure to low SEP leads to poorer dietary outcomes [6].
While I agree that education is an important indicator of SEP, it is insufficient to draw conclusion on the relationship between SEP and diet quality merely based on education. At the minimum, analyses based on other SEP measures such as occupation and income are needed to check the robustness of the results.
We appreciate this comment and agree. Unfortunately the dataset we are employing does not have sufficient data to conduct the analyses you correctly indicate would be needed, and as such we have expanded the limitations section with respect to your point:
Lines 977-9789 Income and occupation are also important indicators of SEP, however; these indicators are not available for both childhood and adulthood in the CFS.
3. Results in Table 2 are very confusing. What are the coefficients for the covariates in the first column of the table label as “Univariates”? Are these bivariate correlations between each covariate and the outcome variable? If so, I suggest use a different table to show the bivariate relationships. Presenting bivariate results and multivariate results in the same table is confusing. Also, results for two models (i.e., one testing for the relationship between childhood SEP and diet quality and the other testing for the relationship between young adulthood SEP and diet quality) are presented together in the column labelled as “Model 1.” It is unclear what the betas for covariates in this column stand for? I suggest present the results in two separate columns.
Thank you for this feedback, we appreciate your suggestions and can see where there may be confusion. We have done two things to address this comment:
1) As suggested, we have included a separate table for bivariate analyses to avoid confusion (now Table 2).
2) We have split Model 1 into two models in two columns in what is now Table 3 – Model 1A and 1B. Model 1A represents the childhood SEP model with covariates, and Model 1B represents the adult SEP model with covariates.
We have also updated the methods and results sections to describe this change. We hope that this clarifies the results.
4. Limitations on the low response rate (20% responded and 48.1% of the responded completed the survey) need to be discussed.
Thanks for this point. We have now addressed this in the limitations section:
Lines 970-974: The survey response rate was low; however, the sample is weighted to be representative of the Canadian urban young adult population and thus generalizability within Canada and potentially to other developed nations with similar demographic and economic characteristics, and systems of governance, is expected to be relatively high.

Round 2
Reviewer 1 Report
I have assessed the revised manuscript and I am satisfied with the authors' responses to my queries. In my opinion this paper is ready to accepted.
Reviewer 2 Report
I appreciate the responses and I don't have further comments.